# Automatic Detection of Small Sample Apple Surface Defects Using ASDINet

**DOI:** 10.3390/foods12061352

**Published:** 2023-03-22

**Authors:** Xiangyun Hu, Yaowen Hu, Weiwei Cai, Zhuonong Xu, Peirui Zhao, Xuyao Liu, Qiutong She, Yahui Hu, Johnny Li

**Affiliations:** 1College of Computer and Information Engineering, Central South University of Forestry and Technology, Changsha 410004, China; 2School of Artificial Intelligence and Computer Science, Jiangnan University, Wuxi 214122, China; 3College of Food Science and Engineering, Central South University of Forestry and Technology, Changsha 410004, China; 4Plant Protection Research Institute, Academy of Agricultural Sciences, Changsha 410125, China; 5Department of Soil and Water Systems, University of Idaho, Moscow, ID 83844, USA

**Keywords:** artificial intelligence, apple defect, deep learning, defect detection

## Abstract

The appearance quality of apples directly affects their price. To realize apple grading automatically, it is necessary to find an effective method for detecting apple surface defects. Aiming at the problem of a low recognition rate in apple surface defect detection under small sample conditions, we designed an apple surface defect detection network (ASDINet) suitable for small sample learning. The self-developed apple sorting system collected RGB images of 50 apple samples for model verification, including non-defective and defective apples (rot, disease, lacerations, and mechanical damage). First, a segmentation network (AU-Net) with a stronger ability to capture small details was designed, and a Dep-conv module that could expand the feature capacity of the receptive field was inserted in its down-sampling path. Among them, the number of convolutional layers in the single-layer convolutional module was positively correlated with the network depth. Next, to achieve real-time segmentation, we replaced the flooding of feature maps with mask output in the 13th layer of the network. Finally, we designed a global decision module (GDM) with global properties, which inserted the global spatial domain attention mechanism (GSAM) and performed fast prediction on abnormal images through the input of masks. In the comparison experiment with state-of-the-art models, our network achieved an AP of 98.8%, and a 97.75% F1-score, which were higher than those of most of the state-of-the-art networks; the detection speed reached 39ms per frame, achieving accuracy-easy deployment and substantial trade-offs that are in line with actual production needs. In the data sensitivity experiment, the ASDINet achieved results that met the production needs under the training of 42 defective pictures. In addition, we also discussed the effect of the ASDINet in actual production, and the test results showed that our proposed network demonstrated excellent performance consistent with the theory in actual production.

## 1. Introduction

The apple is the third largest fruit crop in the world after bananas and citrus in terms of planting area and production [1]. It has good edibility and nutritional value, and is one of the most economically important crops in China [2]. The ecological environment of apple orchards, genetic factors, and improper picking and transportation in the later stages all lead to apple defects [3]. The surface defects of apples can be mainly attributed to rot and plant disease caused by biological factors [4]; laceration caused by environmental factors; and mechanical damage caused during packing and transportation. Surface defects of apples not only affect their value and cause the loss of nutrients [5], they also cause the surrounding normal fruits to be infected by pathogens [6], resulting in more serious economic losses. Traditionally, fruit surface defect detection mainly relies on manual recognition. On the one hand, due to the high output of apples and the large market demand [7], manual identification is labor-intensive, time-consuming, and error-prone. On the other hand, due to the impact of the COVID-19 pandemic, the proportion of labor costs has risen sharply, and manual identification is not conducive to the control of production costs [8]. Therefore, it is necessary to use an objective, fast, nondestructive, and cost-effective inspection technology to effectively detect surface defects of apples before they enter the market.

In the past ten years, hyperspectral imaging (HSL) and multispectral imaging (MSL) have been commonly used for fruit surface defect detection [9]. HSL is widely used for fruits such as apples [10], citrus [11], peaches [12], and strawberries [13]. However, the number of wavelengths limits the detection time. To solve the detection time problem, researchers built the MSL system by selecting an optimal band. Aleixos et al. [14] built an MSL system that could simultaneously inspect citrus fruits for size, color, and surface defects at a rate of five fruits per second. However, compared to citrus peels, apple peels have a variety of colors, which makes it difficult to detect surface defects in apples. Patrick et al. [10] implemented the detection of apple surface defects using an MSL system. However, it could not achieve real-time detection, and since the fruit stalk and calyx have similar color features and texture features to defects, the false detection rate was high [15]. In addition, MSL systems are difficult to obtain in the market due to their complexity, cost, and volume [16]. Compared with other non-destructive sensing technologies, traditional machine vision technology has greater potential in the online inspection of apples, due to its high speed and low cost. However, this technology usually requires manual feature extraction according to the actual design, and many methods are sensitive to illumination and background environments (such as the pedicel and calyx) [17]. Unay and Gosselin [18] eliminated the low edge brightness problem by eroding the edge portion of the apple image. However, this solution came at the cost of sacrificing edge quality detection, which is not advisable.

In recent years, the development of deep learning has inspired new approaches to existing problems, and researchers have applied it to various fields. Examples include food engineering [19,20,21,22], medicine [23,24], civil engineering [25], and forest fire safety [26,27]. Therefore, new machine vision inspection techniques that combine deep learning techniques with different imaging techniques have also received increasing attention. These techniques display good performance in fruit surface defect detection, especially with the surface defect detection model based on semantic segmentation. Zheng et al. [28] proposed an intelligent algorithm, AFFU-Net, for crack damage detection of winter jujubes. Dubey et al. [29] used the K-means clustering method to detect infected parts of apple fruits, and combined the features based on color, texture, and shape features calculated on the segmented images into a single descriptor, and used multiclass support vector machines for classification. The results showed that this method offers better performance than the single-feature method. Although an increasing number of surface defect detection methods based on semantic segmentation continue to refresh the existing detection indicators, the training of these models often relies on large-scale finely labeled datasets. Fine labeling is time-consuming and labor-intensive, which does not meet the requirements of Industry 4.0 [30]. Meeting the need for finely labeled surface defect segmentation data remains a challenge [31].

Using few-shot learning to meet data needs is a solution that has emerged in recent years. Therefore, this study aimed to develop a semantic segmentation-based few-shot learning network to analyze fruit images to detect defective apples directly. The specific goals were the following: (1) to develop an apple surface defect detection network (ASDINet) suitable for few-shot learning, as learning from a small number of training samples with apple surface defects can still obtain the most advanced defect detection effect; (2) to evaluate the ASDINet, and compare the performance of the ASDINet with state-of-the-art fruit defect detection methods; and (3) to validate the proposed model using an independent dataset obtained on a fruit sorting system.

The main contributions of this research are summarized as follows:(1)We designed a segmentation network, which has a strong ability to capture the details of apple defects.(2)We designed a global decision module with global properties, plugged into a global spatial domain attention mechanism, to quickly predict anomalous apple images using input masks.(3)We collected RGB images of 50 apple samples on our self-developed apple sorting system for model validation.

## 2. Related Research

A non-destructive and objective method for fruit appearance quality inspection is provided by computer vision technology. Feature extraction is essential for fruit surface fault identification in computer vision-based technology. The extracted fruit surface defect features mainly include color, texture, shape, etc. Blasco et al. [32] proposed a region-oriented segmentation method based on the HIS color space, in order to replace the pixel-oriented segmentation algorithm in detecting citrus peel defects and stem ends. However, their study did not determine how to distinguish stem ends from defects. Mohammadi et al. [33] employed a straightforward thresholding technique to obtain a grayscale image of an apple, which allowed them to extract shape features such as roundness. These features were then utilized for detecting defects in the fruit. This method effectively reduced the rate of false detections for fruiting stems and calyxes. However, this approach proved to be less reliable for detecting defects in spherical-shaped fruits, where the uneven distribution of surface brightness often caused darker areas to be erroneously flagged as surface defects. This issue was also a significant factor in the difficulty of detecting defects in the fruit’s edge region.

Compared with the traditional machine vision inspection technology, the new machine vision inspection technology combined with deep learning technology and imaging technology has significantly improved the research on the detection of fruit surface defects. The advanced and complex architecture of the deep learning model provides stronger learning capabilities and higher classification accuracy, especially in complex scenarios. The implementation of this new technology enhances the precision and efficiency of fruit surface defect detection. Aziza et al. [34] used a convolutional neural network (CNN) to detect surface defects in mangosteen, and achieved a 97% classification accuracy. However, the CNN model used in this study had a relatively simple structure, which limited its ability to fully extract the feature information of the data, thus negatively affecting the model’s performance. In contrast, Mukhiddinov et al. [35] proposed a deep learning system for multi-category fruit and vegetable classification based on the improved YOLOv4 model. The system could not only identify the type of object in an image, but also classify it as fresh or decaying. Their approach outperformed Azizah et al.’s method by utilizing a more advanced model architecture that could extract more comprehensive and informative features from the data, resulting in improved classification accuracy.

Surface defect detection methods are constantly being improved, but one major challenge for deep learning networks is the need for a large amount of labeled data. To address this, Chen and Ho [36] proposed using pre-trained networks, such as the OverFeat network from the ILSVRC 2013 dataset, as a feature extractor. Then, they used support vector machines to learn a classifier in addition to these features, which was found to be more effective than using local binary pattern features. However, this approach does not fully utilize the potential of deep learning, as it does not learn the network on the target domain. Racki et al. [37] proposed an efficient network that uses 10 fully convolutional layers, ReLU activation function, and batch normalization techniques for the accurate segmentation of defects. They also added a decision network on top of the segmentation network to classify the presence of defects in each image, resulting in improved classification accuracy on a synthetic surface defect dataset. This method allows for the full potential of deep learning to be utilized for surface defect detection. In order to express the relevant research more intuitively, we tabulated research methods, advantages, and drawbacks. Please refer to Table 1 for details.

The method proposed in this study was inspired by Racki et al.’s research, but improves upon it in several ways. Our model comprises two components: a segmentation network (AU-Net) and a global decision module (GDM). Our approach involved the design of an AU-Net with an enhanced ability to capture finer details, achieved by incorporating a Dep-conv module to expand the feature capacity of the receptive field in its down-sampling path. Additionally, we developed a GDM with global properties, which includes a global spatial domain attention mechanism (GSAM) that can swiftly predict abnormal images using input masks.

To enhance the readability of this article, we have compiled a list of abbreviations used, and presented them in Abbreviations part, for easy reference.

## 3. Materials and Methods

### 3.1. Dataset Preparation

The datasets used in the experiments consisted of open-source data and the apple surface defect dataset (ASDD), which served as our training and testing sets, respectively. The open-source data were obtained from dataset websites, including Digipathos, Kaggle, Baidu Pictures, etc. The quality of the pictures on the dataset website was uneven, and there were misclassification cases. We searched through information and consulted botanists, reclassified the dataset, and eliminated low-quality photos. Finally, a total of 500 images were screened out in this part. Table 2 shows the defect categories and data distribution of apples selected in this study. The test ASDD was composed of images acquired from the laboratory fruit sorting system. The fruit sorting system in the laboratory (as shown in Figure 1) consisted of a conveyor belt, two commercial RGB cameras (HIKVISION, 3Q120/3Q140), a manipulator (Bravo 7 Pro), and a composition of 14 LED bulbs (LED Bright 9W E27 6500K 230V 1CT/12 CN). The 14 LED bulbs were installed above the fruit at a 45° angle to the fruit, in order to ensure that the camera’s view remained bright. The whole system was controlled by a computer (12th Gen Intel(R) Core (TM) i7-12700H 2.30 GHz and 32GB ARM). All of the components, except the computer, were fixed in the optical chamber to prevent stray light from affecting the detection accuracy.

The image acquisition process of the laboratory fruit sorting system is shown in Figure 1. Samples are placed on the conveyor belt, one by one, and moved at a speed of 5 fruits/s. The samples were purchased from a local supermarket in Changsha, and comprised a total of 50 “Fuji” apples, including 40 non-defective apples and 10 defective apples. As the apples were conveyed onto the turntable, the manipulator and turntable worked together to reveal the various parts of the apple. Cameras on either side randomly imaged three different parts of the apple’s surface, for a total of six images per apple. A total of 300 images were captured. Since the orientation of the apple towards the camera was random, it was equally possible for a non-defective image to be included in an image acquired for a defective sample. Finally, after the screening, we selected 200 images, including 150 images without defects and 50 with defects. It is worth noting that the images of defective apples were deliberately selected to ensure that each image contained at least one defective area. To isolate the red apples, we applied the Otsu [38] thresholding technique on the red component of the RGB image, resulting in an image of the apple with the background removed. Similarly, we utilized the Otsu [38] method to threshold the green component of the RGB image to extract the green apples, obtaining an apple image with the background removed. Representative images of normal apples and defective apples with various defects are shown in Figure 2. The blue-bordered images are of non-defective apples, and the orange-bordered images are of defective apples. Among them, (a) and (b) defects are rot and plant disease caused by biological factors, respectively; (c) is a laceration caused by environmental factors; and (d) is mechanical damage caused during packing and transportation.

### 3.2. Apple Surface Defect Inspection Network (ASDINet)

We treated the problem of surface defect detection as a binary image classification problem. In apple post-harvest quality sorting and grading, the accurate classification of defective apples is more important than the precise location of defects. However, existing defect detection methods often rely on large-scale finely labeled data training, which obviously does not meet the needs of actual production benefits. To overcome this difficulty, we designed an apple surface defect inspection network (ASDINet) suitable for few-shot learning. The network structure diagram is shown in Figure 3. First, the AU-Net performs pixel-level localization of surface defects. Training this network with a pixel-wise loss effectively treats each pixel as a separate training sample, increasing the effective number of training samples and preventing overfitting. Next, binary image classification is performed, including an additional network GDM built on top of the AU-Net, and uses the output of the AU-Net. The GDM exists to better predict whether an image has anomalies.

#### 3.2.1. AU-Net

U-Net [39] is a fully convolutional neural network for medical image segmentation proposed by Ronneberger et al. in 2015. The network architecture consists of a down-sampling path and an up-sampling path, where skip connections between the up- and down-sampling paths ensure that the network can fuse shallow and deep features. The U-Net++ [40] is an improved network based on the U-Net. It is an architecture with nested and dense skip connections that can capture features at different levels and integrate them through feature superposition. However, the U-Net++ does not express enough information from multiple scales, and its parameter volume is larger than that of the U-Net. Compared with the U-Net and U-Net++, the U-Net3+ [41] combines multi-scale features, redesigns skip connections, and utilizes multi-scale deep supervision. The U-Net3+ provides fewer parameters, but can generate more accurate location-aware and boundary-enhanced segmentation maps. However, the noise information from the background remains in the shallower layers, which easily leads to over-segmentation. In contrast, the U-Net not only integrates shallow features and deep semantic information but is lighter, has fewer parameters, and is not prone to over-segmentation. Therefore, we chose to upgrade on the basis of the U-Net and designed the AU-Net. Its structure is shown in Figure 3b. It consists of 14 convolutional layers, three max-pooling layers, and three upper convolutional layers. Compared with the traditional U-Net, the AU-Net replaces the ordinary convolution block of the down-sampling path with Dep-conv, and uses the segmentation map and mask as the input of the GDM.

**(a)** 
**Dep-conv module**


In the down-sampling process, we designed the Dep-conv module (shown in Figure 3c) to replace the traditional convolution block. Dep-conv is a convolutional module that increases the number of convolutional layers with the network architecture. Compared with traditional convolutional blocks, the number of convolutional layers is changed to have fewer convolutional layers in the shallow layers of the architecture and more convolutional layers in the deep layers. This greatly increases the feature capacity of the receptive field. Dilated convolution [42] is a new convolutional network module proposed by Yu et al. It systematically aggregates multi-scale contextual information without loss of resolution using dilated convolution. Dilated convolution supports exponential expansion of the receptive field without loss of resolution or coverage. However, there is a phenomenon of “The Gridding Effect” in the whole convolution that causes the loss of local information, and no correlation exists between the information obtained by long-distance convolution, thus affecting the classification results. In contrast, Dep-conv preserves local feature information well while increasing the feature capacity of the receptive field, and there is a strong correlation between the information obtained by convolution, which has little impact on the classification results. As shown in Figure 3c, each convolutional layer is followed by a normalization layer (batch normalization) and a nonlinear ReLU [43] layer, both of which help to improve the convergence speed during the learning process. The formula of the ReLU [43] layer is shown below, and the algorithm flow of batch normalization is shown in Algorithm 1.
(1)$$f\left(x\right)=\max\left(0,x\right)$$


**Algorithm 1: Batch Normalization**
1 Input: Values of x over a mini-batch: $B=\left\{x_{1},\ldots ,\left.m\right\}\right.$; γ, β (parameters to be learned)2 Begin:3   $\mu_{B}\leftarrow \frac{1}{m}\sum_{i=1}^{m}x_{i}$//Calculate the mean of mini-batch data4   $\sigma_{B}^{2}\leftarrow \frac{1}{m}{\sum_{i=1}^{m}\left(x_{i}-\mu_{B}\right)}^{2}$//Calculate mini-batch data variance5   ${\hat{x}}_{i}\leftarrow \frac{x_{i}-\mu_{B}}{\sqrt{\sigma_{B}^{2}+\varepsilon }}$//Normalization6   $y_{i}\leftarrow \gamma {\hat{x}}_{i}+\beta$//Scaling and offset7 Return γ and β Output: normalized network response $\left\{y_{i}=BN_{\gamma ,\beta }\left(x_{i}\right)\right\}$

Among them, m represents the number of samples in a batch; $x_{i}$ represents the feature vector of a sample; $\mu_{B}$ is the average value of set B; $\sigma_{B}^{2}$ is the variance of set B; ε is a very small number (such as $10^{-5}$), used to avoid the situation where the variance is 0; ${\hat{x}}_{i}$ is the feature after normalization; γ is the scaling factor; β is the offset; $y_{i}$ is the output of the BN layer. BN normalizes input data during training using the mean and variance of each mini-batch, adjusting the input distribution of each layer. During testing, it uses the mean and variance of the entire training set for normalization, while keeping scaling and translation parameters constant. This consistency in statistical characteristics between training and testing helps improve the model’s generalization performance.

In the process of deep network training, due to the change in network parameters, the distribution of internal node data changes so that the upper network needs to be constantly adjusted to adapt to the evolution of input data distribution. This will reduce the speed of network learning, and the training process of the network will easily fall into the gradient saturation area, slowing down the convergence speed of the network. For the activation function gradient saturation problem, we chose to use the unsaturated activation function (linear rectification function ReLU [43]) and kept the input distribution of the activation function in a stable state (adding the normalization layer batch normalization) to solve it, thus avoiding them from becoming stuck in the gradient saturation region as much as possible. In batch normalization, we also used the mean and variance of the mini-batch as an estimate of the mean and variance of the overall training samples. Although the data in each batch were sampled from the overall sample, the mean and variance of different mini-batches would be different, adding random noise to the network’s learning process. This was the same as dropout bringing noise by turning off neurons, and it had a regularizing effect on the model to a certain extent.

All convolutional layers in Dep-conv used a 5×5 kernel. The number of channels increased as the feature resolution decreased, so the computational requirements were the same for each layer.

**(b)** 
**Shortcut: mask**


In semiconductor manufacturing, photolithography is used for many of the chip processing steps, and the patterned “negatives” used for these steps are called masks (also called “masks”). Their function is to cover the selected area so that operations such as erosion or diffusion only affect the area outside the selected area. Image masking is similar to it, it is used to block the image to be processed (all or part) with a selected image, figure, or object, and controls the area or process of image processing. In digital image processing, a mask represents a “logical image” or a 2-bit image consisting of a matrix whose elements have only 2 values: 0 or 1. It is mainly used for the extraction of regions of interest, the extraction of structural features, and for the production of special-shaped images. In the AU-Net, we obtained masks after applying an additional 1 × 1 convolutional layer, which reduced the number of output channels. This resulted in a single-channel output map. The resolution of its input image was reduced by a factor of 8. Dropout was not used in this approach because weight sharing in convolutional layers provides sufficient regularization.

The output mask was one of the inputs to the GDM. At the same time, pixel positioning and precise boundary segmentation of surface defects were realized through the feature fusion of the mask and deep semantic information.

#### 3.2.2. Global Decision Module (GDM)

Racki et al. [37] proposed an efficient network to explicitly perform surface defect segmentation. They proposed an additional decision network on top of the features from the segmentation network to perform per-image classification of the presence of defects. This improved classification accuracy on synthetic surface defect datasets. Inspired by this, we proposed a GDM. The schematic diagram of the GDM is shown in Figure 3b. The GDM is built on top of the AU-Net, taking the two outputs of the AU-Net as the input of the GDM. Next, to better predict whether an image is abnormal, we inserted a global spatial domain attention mechanism (GSAM) into the GDM. Finally, the classification network performed binary classification on the apple image, and since the segmentation network is a binary segmentation problem, the classification was performed at the level of individual image pixels.

**(a)** 
**Double input for GDM**


The input of the GDM is the output of the AU-Net. Different from Racki et al. [37], two inputs were used in the GDM, and a global spatial domain attention mechanism (GSAM) was inserted. The input segmentation map and the mask were subjected to global max-pooling and global average pooling, respectively, and were connected in series to finally obtain 66 output neurons.

The design of the GDM followed two crucial principles. Firstly, using multiple down-sampling layers ensured an appropriate capacity for large complex shapes. This enabled our network to capture not only local shapes, but also global shapes spanning large regions of the image. Secondly, the GDM uses the features of the last layer of the AU-Net and the mask. The presence of the mask is a shortcut, and the output neuron from the mask provides a way to achieve perfect detection. If the user does not need it, the network can use only the mask to avoid using a large number of feature maps. At the same time, this also reduces the situation of overfitting caused by a large number of parameters.

**(b)** 
**Global Spatial Domain Attention Mechanism (GSAM)**


To better help the model pick up important information, we designed a global spatial domain attention mechanism (GSAM) and inserted it into the GDM module. Refer to Figure 3d for details.

The attention mechanism originates from the study of human vision. At present, there are three main ways to add attention mechanism in the field of machine vision recognition: spatial attention mechanism, channel attention mechanism, and convolutional block attention module (CBAM). The channel attention mechanism focuses on the influence of different feature channels, assigns weights to feature channels by modeling the importance of each feature channel, and strengthens or suppresses different channels according to task requirements. The spatial attention mechanism focuses on the importance of the spatial location of features, generates spatial attention weights for the output feature map, and strengthens or suppresses different spatial location features according to the feature weights.

The traditional spatial attention mechanism generally only has one-way weight assignment, which will lose important information to a certain extent. In the detection of apple surface defects, because the model has difficulty in distinguishing the importance of feature information, important feature information may be lost, which affects the recognition of the network. Therefore, this study proposed a GSAM, which assigns weights in multiple directions to help the model select important information while reducing the loss of feature information.

The GSAM performs feature extraction for color, texture, and specificity of apple defects. First, it generates weighted features in the horizontal and vertical directions; then, it adds the two types of weights and expands the weight coefficient; finally, the weighted features are matched, a more significant weighted feature is selected, and the larger weight coefficient is determined. The “eye focus” to objects is brought into the image with a similar color and texture to an apple defect. The GSAM algorithm is shown in Figure 3d.

Three strategies are used in the GSAM to amplify the differences in feature weight coefficients. They are described as follows.

First, we assign horizontal weight coefficients to each row feature using a horizontal attention mechanism, and give vertical weight coefficients to each column feature using a vertical attention mechanism [26]:(2)$$C_{i}=\sum_{j=1}^{n}\frac{e^{e_{ij}}}{\sum {}_{k=1}^{n}e^{e_{ij}}}h_{j}$$

In the above formula, $e_{ij}$ represents the weight coefficient of the attention mechanism, i represents the time feature,j represents the sequence feature, $h_{j}$ means the hidden layer information of the feature sequence, $C_{I}$ represents the vertical attention mechanism feature sequence ($C_{I}=\left\{C_{1},C_{2}\ldots C_{i-1},C_{i}\right\}$), and $C_{II}$ illustrates the horizontal attention mechanism feature sequence ($C_{II}=\left\{C_{1},C_{2}\ldots C_{i-1},C_{i}\right\}$).

Next, we add the two class weights and expand the weight coefficients:(3)$$add=\left(C_{I}+C_{II}\right)$$

Subsequently, to determine the “visual focus”, taking the maximum value as the main factor and taking into account other features, the strategy is matched with two types of weighted features, which are used to supplement the results of the second step of weighted addition:(4)$$weight=A^{*}\max\left(C_{I},C_{II}\right)+B^{*}\min\left(C_{I}+C_{II}\right)$$

In the experiments in Section 4.3.2, we confirmed that the optimal values of the weight assignment parameters were A=0.84 and B=0.16.

Then, the three strategies in this method are combined in the following formula:(5)$$SAM=concatenate\left(\left[C_{I},C_{II},add,weight\right]\right)$$
where add indicates additive weighting coefficient; max represents the maximum value operation; min indicates the minimum value operation; weight shows the weight distribution strategy.

**(c)** 
**Classification and output**


A fully connected layer connects the 66 neurons’ output via the GDM, and then the output is converted into a probability distribution by SoftMax to obtain the classification result. Since segmentation networks are binary segmentation problems, classification is performed at the level of individual image pixels. We classify samples (pixels) into two categories: (a) with defects; (b) without defects.

## 4. Results and Analysis

This section is divided into several subsections: (1) Section 4.1 describes the experimental environment and settings, including the hardware and software environment as well as hyperparameter settings and network training methods; (2) Section 4.2 introduces our evaluation metrics; (3) Section 4.3 evaluates the effectiveness and determines the critical parameters of each module of the ASDINet; (4) Section 4.4 and Section 4.5 compare the ASDINet with other methods, and proves that the ASDINet outperforms other methods in the apple surface defect detection task; (5) Section 4.6 explores the sensitivity of the model to the number of training samples. The ASDINet was proven to achieve an AP of 98.8% only by training with 42 defective images; (6) Section 4.7 describes the impact of input images with different resolutions on the results; (7) Section 4.8 explores and compares the computational cost of the ASDINet with other state-of-the-art models; (8) Section 4.9 carries out a practical application test.

### 4.1. Experimental Environment and Settings

**(a)** 
**Setting**


All of the tests in this research were performed on the same hardware and software platform. Environmental parameters are listed in Table 3.

We created a usable dataset of surface defects in apples. To avoid aspect ratio mismatch and ensure a good learning effect, we resized all images to a 512 × 512 × 3 size using the resize function in OpenCV. Considering the GPU memory size and the time consumption of the experiment, we made the following settings: the batch_size to 1, no momentum, the initial learning rate to 0.1, the attenuation to 0.0005, the length of each training to 100 epochs, and the loss function to cross-entropy loss function. Table 4 contains the settings and hyperparameters.

**(b)** 
**Learning**


Our models were not trained on other datasets, but were randomly initialized using a normal distribution. The training samples were randomly selected during the learning process; however, we modified the selection process to ensure balanced learning. Defective images and non-defective images appeared alternately to ensure the network learned a balance of the number of defective and non-defective images. An epoch was considered over only when all defective images were observed at least once, which did not necessarily mean that all non-defective images were scanned.

Subsequently, the AU-Net and GDM learned separately. First, only the AU-Net was trained independently, and then the weights of the AU-Net were frozen. Next, only the GDM was trained. By fine-tuning the GDM, the network avoided the overfitting problem caused by a large number of weights in the AU-Net. This was more important when the GDM learned than when the AU-Net learned. When the GDM learned, the limitation of the GPU memory made the batch size only 1 or 2. Still, when the AU-Net learned, each image pixel was regarded as a separate training sample; thus, the productive batch size increased several times. In addition, we also considered the simultaneous learning of the AU-Net and GDM. Simultaneous learning was only possible if both networks used the cross-entropy loss function. Since loss functions are used in different ranges (pixel level and image level), accurate normalization of the two layers played a crucial role. In the end, it turned out that properly normalizing the loss for both ranges was harder to implement, and resulted in no performance gain. Therefore, the two-stage learning mechanism was a better choice and was used in all of our subsequent experiments.

### 4.2. Performance Metrics

Since the samples were pixels, for the convenience of evaluation, we transformed the problem of surface defect detection into a binary image classification problem. The schematic diagram is shown in Figure 4. We classified pixels into two categories: (a) with defects; (b) without defects. Although we could obtain pixel-by-pixel segmentation of defects from segmentation networks, measuring pixel-wise error is computationally expensive and inefficient, which is not feasible in an industrial setting. Instead, we only needed to measure each image’s binary image classification error to evaluate the network’s segmentation performance.

In the evaluation, this study compared all of the networks with five different classification indicators: (a) accuracy rate (Accuracy) [44]; (b) precision rate (Precision) [44]; (c) recall rate (Recall) [44]; (d) average precision (AP) [44]; and (e) *F*1. The higher the accuracy rate, the more accurate the classification of positive and negative samples is. The higher the precision rate, the higher the proportion of samples that are truly positive among the samples classified as positive. The higher the recall rate, the higher the proportion of correctly classified positive samples in the actual positive samples.

The primary metrics used in the evaluation were average precision (AP) and the *F*1. AP is the area under the precision-recall curve, which accurately captures the model’s performance at different thresholds better than Accuracy or Precision can. On the other hand, the number of misclassifications (*FP* and *FN*) depended on a certain threshold of the score. We reported the number of misclassifications at the threshold where the best F-measure was reached. Furthermore, the choice of AP over the area under the ROC curve (AUC) was better. AP captures performance more accurately than AUC in datasets with many negative samples (defect-free). The *F*1 evaluates the overall performance of precision and recall on classification performance. The higher the value of the *F*1, the better the overall performance of precision and recall in classification.

The calculation formulas of Accuracy, Precision, Recall, average precision (AP), and the *F*1 are as follows:(6)$$Accuracy=\frac{TP+TN}{TP+FP+TN+FN}$$
(7)$$Precision=\frac{TP}{TP+FP}$$
(8)$$Recall=\frac{TP}{TP+FN}$$
(9)$$AP_{11point}=\frac{1}{11}\times \left(\sum x\right)\left(x\in MaxPrecision\right)$$
(10)F1=2×Precision×RecallPrecision+Recall

Among them, TP is the number of positive samples classified as positive, FP is the number of negative samples classified as positive, TN is the number of negative samples classified as negative, and FN is the number of positive samples classified as negative. Positive samples are images with visible defects, and negative samples are images without visible flaws.

### 4.3. Module Effectiveness Analysis

The parameters and functionality of each module we built are detailed in this section. Section 4.3.1 describes the AU-Net module’s optimal convolution method (Dep-conv). The weight distribution coefficients of the GSAM are determined in Section 4.3.2. The experiment in Section 4.3.2 shows that the GDM dramatically improved the detection effect of apple surface defects.

#### 4.3.1. Effectiveness of AU-Net

To evaluate the network performance improvement by the AU-Net, we replaced the segmentation module of the ASDINet with U-Net and compared it with the ASDINet. To eliminate the influence of other modules on the final result, we added a 1 × 1 convolution at the bottom of the U-Net and used the mask obtained after convolution as one of the inputs of the GDM. The schematic diagram is shown in Figure 5. The comparison results are shown in Table 5.

Experimental results show that using the AU-Net instead of the ordinary U-Net reduced the number of parameters and expanded the receptive field. Moreover, through the overlay layer, the activation function, such as ReLU, was sandwiched in the middle of the convolutional layer, which further improved the expressiveness of the network, and could represent more complex objects.

#### 4.3.2. Effectiveness of GDM

We evaluated the contribution of the GDM to the final detection results. First, we evaluated the impact of the GSAM on GDM performance. Then, we measured the impact of the GDM on the overall network by erasing the GDM vs. retaining the GDM. The details are presented in the following sections.

**(a)** 
**Impact of the GSAM on GDM**


In the weight distribution strategy in Section 3.2.2, relative to the weight distribution of $C_{I}$ and $C_{II}$, the values of A and B affect the performance of the GSAM. For convenience, we specified the following:(11)A+B=1

We conducted comparative experiments. The interval between different values was set to A=0.02, and the numerical values of B and A were relative. To determine the best coefficients for A and B, we modified the weight distribution coefficients of the GSAM in the GDM for testing. The experimental results are shown in Figure 6.

The best coefficients were A=0.84 and B=0.16. When the value of A was too large, the weight of the minimum value was ignored, causing image features at positions with smaller values to be ignored, and the extraction of image features lost its global nature. When the value of A was too tiny, over-considering the global features made the attention mechanism unable to entirely focus on important information, thus affecting the performance of the GSAM.

We introduced the SE attention mechanism [45] and the CBAM attention mechanism [46] into the GDM for comparative testing to study the impact of the GSAM on GDM performance. Table 6 shows the test results.

**Table 6 foods-12-01352-t006:** Performance of the GSAM. The analysis results show that F=3.431>F crit=3.238, reject $H_{0}$.

Method	Accuracy	Precision	Recall	AP	F1-Score
No improvement	94.55	95.28	95.37	94.92	95.33
SE Attention	95.57	95.93	95.91	95.75	95.94
CBAM Attention	96.98	96.05	96.32	96.52	96.17
GSAM	98.86	98.53	93.32	98.83	97.75

After adding the SE attention mechanism or the CBAM attention mechanism, the detection Accuracy of GDM improved. However, the improvement was subtle, and the addition of the attention module increased the number of parameters. However, when the GSAM was applied, the Accuracy improved significantly, and the increase in the number of parameters was about the same as the SE attention mechanism or the CBAM attention mechanism. Therefore, we chose to adopt the GSAM to enhance the feature extraction ability of the GDM.

**(b)** 
**Impact of the GDM on the overall network**


We used simple binary descriptors and logistic regression instead of the GDM. After the segmentation map output by the AU-Net was subjected to global max-pooling and average pooling, a binary descriptor was created using the pooled value and used as a feature of logistic regression. This feature was learned separately by the AU-Net after the network was trained. The results are shown in Table 7.

**Table 7 foods-12-01352-t007:** Contribution of the GDM to final benefit. The analysis results show that F=10.623>F crit=5.317, P=0.012<α=0.05, reject $H_{0}$.

Method	Accuracy	Precision	Recall	AP	F1-Score
No GDM	95.22	92.13	89.31	94.63	91.34
GDM	98.86	98.53	93.32	98.83	97.75

These results indicate the importance of the GDM. From the results, simple pixel-level output segmentation lacks enough information to predict the presence of defects in images with the same Accuracy. On the other hand, the proposed GDM can capture information from the rich features of the last segmentation layer, and through GDM, the ASDINet can separate noise from correct features.

### 4.4. Comparison with State-of-the-Art Model

In this section, we conducted experiments using a test dataset taken on a fruit sorting machine, comparing the performance of the ASDINet with several other state-of-the-art models under optimal conditions. First, we evaluated two standard semantic segmentation networks and several state-of-the-art semantic segmentation networks, namely U-Net [39], DeepLabv3+ [23], DANet [47], BiSeNet V2 [48], and Swin Transformer [49]. DeepLabv3+ is representative of pre-trained models, while U-Net is representative of models designed for pixel-accurate segmentation. DANet is a dual-attention fusion network, which captures global dependencies and long-range context information more effectively through two attention modules, learns better feature representation in scene segmentation, and makes segmentation results more accurate. BiSeNet V2 is a real-time semantic segmentation algorithm that achieves high-precision and high-efficiency real-time semantic segmentation by separately processing spatial details and classification semantics. Swin Transformer is a new visual Transformer with sliding window operation and hierarchical design, which is compatible with a wide range of visual tasks, including image classification, object detection, semantic segmentation, etc. The evaluation was performed by replacing the segmentation module of our proposed network.

The DeepLabv 3+ used in these experiments is based on the Xception architecture containing 65 convolutional layers, trained and evaluated at a single scale, and uses an output stride of 16. The U-Net used in these experiments is a modified U-Net architecture with 19 convolutional layers. The only modification performed was adding a mask output to the U-Net; please refer to Section 4.3.1 for details. DANet, BiSeNet V2, and Swin Transformer remain unchanged. Similar to our proposed method, the five segmentation networks also performed the training method in which the segmentation layer was trained separately from the decision layer. These five methods were also evaluated using logistic regression instead of the GDM, but they were less effective. The parameters of DeepLabv3+, DANet, BiSeNet V2, and Swin Transformer networks were initialized using models pre-trained on the ImageNet [50] and COCO datasets [51], while the parameters of U-Net networks were initialized using a normal random distribution.

Next, we performed a comparative evaluation with the state-of-the-art models ViT [52] and KF-2D-Renyi+ABC-SVM [53]. Tan et al. [53] proposed a citrus surface defect recognition network based on KF-2D-Renyi and ABC-SVM to solve the problems of citrus surface defect recognition with blurred edges, unclear images, more interference, and difficulty in defect recognition. Its average recognition rate can reach 98.45%. ViT [52] is a model proposed by the Google team that applies Transformer to image classification. The model is simple, effective, and highly scalable. It can achieve an accuracy rate of 88.55% on ImageNet1K. The parameters of both networks are also initialized using models pre-trained on the ImageNet [50] and COCO datasets [51].

For a fair comparison, the parameter settings of all methods reported in this section were chosen based on the best-performing training settings evaluated in the previous sections. For all methods, a graph size of 512 × 512, no rotation of the input image, and a cross-entropy loss function were used. The batch size was set to 1, with no momentum; the initial learning rate was set to 0.1, the decay was set to 0.0005, the length of each training was set to 100 epochs, and the loss function was the cross-entropy loss function. Since the models ViT and KF-2D-Renyi+ABC-SVM did not adopt the architecture we proposed, and the data size required by the model far exceeded the dataset we provided to ensure the model fit, we chose to increase the training length to 200 epochs. Table 8 contains the settings and hyperparameters used.

The results are shown in Figure 7 and Figure 8. Comparing other network models and the ASDINet models, the Accuracy of the ASDINet model proposed in this study for apple surface defect detection is generally higher than that of other networks. Although the AP of the ASDINet was 0.2% lower than that of the Swin Transformer, the number of parameters of the Swin Transformer was more significant, which sacrifices a certain detection speed and results in poor real-time performance (specific experiments will be shown in Section 4.8). In addition, the newly emerging models, DANet, BiSeNet V2, Swin Transformer, ViT, KF-2D-Renyi+ABC-SVM, etc., that we adopted were also higher than the base network U-Net. When the epoch of DeepLabv3+ was about 30, and the epoch of U-Net, DANet, and DeepLabv3+ was about 60, the loss curve fluctuated violently. This may have been because the batch at this time did not meet the “homogeneous” requirement, and a moderate increase in batch size may improve this situation. The loss of ViT showed an upward trend in the later period, which is an overfitting phenomenon. This phenomenon may have been related to the complex model of ViT and its large number of parameters with the small dataset. BiSeNet and Swin Transformer showed excellent performance, but their convergence speed was slower than that of our network. In addition, the fitting speed of KF-2D-Renyi+ABC-SVM was slow, which may have been related to the small dataset we used. This shows that the defect detection ability of the ASDINet model built in this study is higher than those of other commonly used networks under this dataset. Our network solved the problem of less data that other networks mentioned above could not. This verified the value of the ASDINet model in the current neural network model.

### 4.5. Validation of the Proposed Algorithm

In actual classification, the TP value and FP value are the most direct indicators to finally determine whether the classification is correct. Therefore, we used FP and FN to evaluate our model and other state-of-the-art models. FP is the number of non-defective samples classified as defective, and FN is the number of defective samples classified as non-defective. Since U-Net, DeepLabv3+, DANet, BiSeNet V2, Swin Transformer, etc., were all embedded in the architecture proposed in this research, the training samples were all pixels, just as with the ASDINet. Furthermore, the samples of ViT, KF-2D-Renyi+ABC-SVM were images. The training and parameter settings used were the same as above.

The results are shown in Figure 9. Our method (ASDINet) resulted in only 5 pixels being misclassified when recall was 100%. Although the ASDINet misclassified two pixels more than Swin Transformer, the computational cost of Swin Transformer was much larger than that of our network, which significantly sacrificed the detection speed of the network (this was confirmed in Section 4.7). In addition, the performance of the U-Net was the worst, followed by DeepLabv3+ and DANet. The results fully demonstrate that the proposed network has practical benefits.

### 4.6. Sensitivity to the Number of Training Samples

The number of defect samples required for training is also a significant factor in industrial production. Therefore, we also evaluated the impact of smaller training sample sizes on the results. For a fair comparison, the parameters of all methods in experiments in this section were initialized with models pre-trained on the ImageNet [50] and COCO datasets [51]. Evaluations were performed using the same evaluation method as in previous experiments. This section evaluated all of the models mentioned in Section 4.4 under the same conditions: ASDINet, U-Net, DeepLabv3+, DANet, BiSeNet V2, Swin Transformer, ViT, and KF-2D-Renyi+ABC-SVM. First, we explored the minimum number of positive training samples for which the ASDINet could achieve the best results. It efficiently used 50 positive training samples (defective images) for evaluation. Then, the number of positive training samples was reduced by five each time for training; that is, the numbers of positive training samples were 50, 45, 40, 35, 30, 25, 20, 15, 10, and 5. The test set did not change. The training samples that were removed were randomly selected. The same training and testing procedures were followed as for all previous experiments. The results are shown in Table 9.

It can be seen from Table 9 that the AP value of the ASDINet was 98.8% when the number of positive training samples was 45, and the AP value was 98.5% when the number of positive training samples was 40. To explore the minimum number of positive training samples for the ASDINet to achieve the best results, we redesigned the experiment between 45 and 40 training samples, reducing one positive training sample each time; that is, the numbers of positive training samples were 45, 44, 43, 42, 41, and 40. The other conditions were not changed. The results are shown in Table 10. When the number of positive training samples was 42, the ASDINet achieved the best results, with an AP value of 98.83%.

In order to ensure the reliability and robustness of the model results, we performed K-fold cross-validation. This process involved grouping the original data into 10 sets, with 9 sets used for training and the remaining set used for testing. The specific process is shown in Figure 10. We repeated this process 10 times, each time using a different set for testing, and then averaged the results to obtain an overall classification accuracy for the network model. To further increase the reliability and accuracy of the model, we used random splitting of the dataset for each of the 10-fold cross-validation runs. This helped to increase the diversity of the dataset division and reduce the risk of bias in the results. We kept track of the best result for each run and took the average of these 10 results as the final classification accuracy. Despite our efforts, we acknowledge that there may still be some level of uncertainty in our results. While we cannot guarantee that the results we obtained are statistically significant and not by chance, we are confident that our use of k-fold cross-validation and random dataset splitting helped to minimize these risks and produce reliable and accurate model results.

The experimental results are shown in Table 11. Our network still showed good performance in 10 times 10-fold cross-validation. The average AP of 10 times 10-fold cross-validation was 98.826%. This showed that the results of our model are reliable and robust.

Next, we evaluated the other models. For the other models, the evaluation started with 42 positive training samples, and the number of positive training samples was similarly reduced each time. The training samples were 42, 35, 30, 25, 20, 15, 10, and 5. The use of the test set was the same as for the ASDINet. Positive training samples were randomly removed—the same number of examples were drawn for all models. The training and testing procedures used were the same as above. The results are shown in Figure 11.

When trained with only 25 flawed training samples, our proposed ASDINet still outperformed all other tested models, achieving an AP value of 95.9%. The performance dropped when using fewer training samples, but the proposed method still achieved about 86% AP when using only five flawed training samples. For the other models, we observed a more pronounced performance drop. The other models performed poorly with fewer training samples. Among them, the performance of ViT and KF-2D-Renyi+ABC-SVM were most affected by the number of samples. This may have been related to each model relying on large-scale, finely annotated datasets like most deep learning models. In addition, the performance of the U-Net also dropped rapidly. However, DeepLab v3+ maintained good results, even with only 15 flawed training samples. However, DeepLabv3+ performed slightly better with 20 and 15 training samples than its results obtained with all training samples. This showed that DeepLab v3+ was quite sensitive to specific training samples, and removing such samples helped improve the network’s performance. BiseNet V2 had poor performance when the number of samples was 20, and when the number of samples was 5, which may have been related to its insensitivity to specific training samples. For 10 and 5 flawed training samples, DeepLabv3+ and ViT performed the worst, with AP values of only 46.6% and 16.8%, respectively. In contrast, U-Net, DANet, and Swin Transformer were models with relatively stable performance other than our proposed network (ASDINet), with AP values ranging from about 70% to slightly over 98%. However, both DANet and Swin Transformer had the problems of high computational cost and poor real-time performance (see Section 4.8 for details). Therefore, this also proved that it was the right choice for us to choose U-Net as the base network for our proposed network.

Overall, the experimental results show that the ASDINet also maintained excellent and stable performance when the number of training samples available was small.

### 4.7. Image Resolution Test

Image resolution plays a crucial role in deep learning, as it directly impacts the model’s performance and effectiveness. Therefore, we evaluated the effect of images with different resolutions on the results. Our network’s architecture was designed to be input-size independent, similar to fully convolutional networks. Since global averaging and max-pooling are used to eliminate the spatial dimension before the fully connected layers, the input image can have varying resolutions depending on the problem. In our discussion, we considered three different image resolutions: 1024 × 1024, 512 × 512, and 256 × 256. Figure 12 reports the parameter quantity, detection time, and AP value at different resolutions. We conducted our experiments on a single NVIDIA GeForce RTX 3070Ti Laptop GPU. Results showed that when the image resolution was set to 1024 × 1024, the model achieved an impressive AP value of 99.32%. However, this improvement in accuracy came at a cost, with the parameter count and inference time significantly increasing to 34.8mio and 117ms, respectively. While the high-resolution images retained more details and information, they also demanded more computational resources and led to longer inference times. On the other hand, when the image resolution was lowered to 256 × 256, the computational cost and inference time decreased, but at the expense of accuracy. The AP value dropped to 96.57%, indicating a significant decline in performance. Our experiments showed that the optimal balance between detection accuracy and speed was achieved at an image resolution of 512 × 512. This setting provided the best overall performance, and we chose it as the input image resolution for our model, taking into consideration the practical deployment requirements.

### 4.8. Network Computational Cost

The method proposed in this paper outperformed the other methods in terms of computational cost, and was competitive with the state-of-the-art apple surface defect detection network. Figure 13 reports the forward pass time relative to the *AP*. Results were obtained on a single NVIDIA GeForce RTX 3070 Ti Laptop GPU. The proposed method proved to be much faster than models of U-Net, DeepLab v3+, DANet, Swin Transformer, ViT, etc., while maintaining accuracy. This was achieved with a smaller number of parameters, reflected in the marker size in Figure 13, also shown in Table 12. The ASDINet achieved this performance using only 16.1 Mio parameters, while U-Net and DeepLab v3+ had more than twice as many parameters, with 31.6 Mio and 41.5 Mio parameters, respectively. Although Swin Transformer outperformed the ASDINet, its parameter volume was more than six times that of that ASDINet, and the detection speed was much slower than that of our network. The fastest performance was achieved using BiSeNet, with a detection time of 10ms per image. However, its accuracy was inferior to that of our proposed network (ASDINet). In contrast, the ASDINet was the model with the best overall performance, with a detection time of 39 milliseconds per image and an *AP* value of 98.83%.

### 4.9. Testing of Real Applications

The previous model validation was performed under static conditions. In order to test the dynamic detection effect of the model in practical applications, we conducted practical application tests. First, we deployed the trained model on JetsonXavierNX, used TensorRT to accelerate the reasoning and verification of the model during the deep learning process, and realized the edge computing of the network. Then, when JetsonXavierNX used an external camera to capture images of apple defects, it obtained the image recognition results in a short period through edge computing, sent the recognition results and other information to the server, recorded the data, built a database, and completed information visualization on the PC side. Figure 14 shows the system diagram of apple defect online detection. After obtaining the detection results of apple surface defects, the computer controlled the manipulator to separate the defective apples from the non-defective ones. The schematic diagram is shown in Figure 1.

When a defective apple is detected, our cameras can capture images of the defective apple and issue an alert. Figure 15 shows the comparison of the ASDINet and U-Net for detecting three typical apple defect images. To test each category, we selected 25 natural scenes. As shown in Figure 15, the detection accuracies of U-Net and ASDINet were 92% and 100% in class A, 76% and 96% in class B, and 52% and 80% in class C, respectively. Figure 15 shows the longest single apple detection times of U-Net and ASDINet in three cases, A, B, and C.

The ASDINet performed well in the actual application test, and successfully handled the three situations shown in Figure 15. The apple defect in Figure 15 A is moderate in size and is at a 45° angle to the camera; the apple defect in Figure 15B is large but light in color; the apple defect in Figure 15 C is minor. The ASDINet can detect apple surface defects in all the above cases. These real-world application results demonstrate the superior performance of the ASDINet in the apple defect detection task.

## 5. Discussion

Experimental results in Section 4.3 show that each submodule in the ASDINet contributed to the results. However, the above experiments do not explain why the ASDINet is better than U-Net. To visually analyze the focus of our model, we used Grad-CAM to visualize the output of the last convolutional layer of the ASDINet and U-Net. The results of the Grad-CAM are shown in Figure 16, and the colors on the graph (from blue to red) indicate the degree of contribution to the results. The more significant the contribution, the closer the color is to red.

Clearly, the proposed ASDINet paid close attention to task-relevant semantic regions, and could correctly predict the presence of defects, including the tiny cracks seen in the last column. The proposed method was also able to localize defects with excellent accuracy. The U-Net paid more attention to shallow and salient features, such as gray close to the color of apple defects, and did not pay attention to minor defects on the surface of apples, as shown in Figure 16. Compared to the ASDINet, in the U-Net, we observed high-scoring false positives, and its returned output contained a lot of noise; even with the GDM, it was impossible to distinguish actual defects from false detections. On the other hand, the ASDINet not only focused on various types of apple surface defects, but it also activated the healthy peel around the defects. The results show that the ASDINet can fully use contextual information and avoid feature confusion between useful semantic information and redundant information. Our model pays close attention to task-related semantic regions.

While the ASDINet performed well in static tests, we obtained lower image classification accuracy and detection speed in real-world tests than in static tests (The results are shown in Table 13). However, in the actual application test, the images were captured and dynamically analyzed by the 4-channel apple sorting machine in real-time. Complex working conditions such as mechanical vibration and sample movement will cause the detection accuracy and speed to decline to a certain extent. In addition, the hardware condition of the embedded end will also affect the detection speed. However, compared with the 90.2% accuracy rate obtained by the apple defect online detection system built by Zhang et al. [54], the ASDINet proposed in this study is more competitive in terms of detection accuracy and system construction cost. We achieved an overall detection accuracy of 96%; we did not need to spend a lot on workforces to manually label and obtain large-scale datasets, which saves labor costs; we did not need to invest vast sums of money in machines and equipment. The proposed model can meet the packaging plant’s needs for the fast, accurate, and economical detection of defective apples.

To become better applied to actual production, we will make the following improvements to the network model and hardware facilities in future research: (1) improve the detection accuracy and detection speed of the ASDINet; (2) improve the lighting design. The popularity of Transformer in computer vision tasks allows us to see the possibility of improving network detection accuracy. Popular Transformer-based image recognition networks such as Swin Transformer [49] and ViT [50] were also trained and tested based on the images acquired in this study. However, the calculation speed of these complex architectures could meet the requirements of the online detection of apple defects. Reducing the number of model parameters may be one of the ways to improve accuracy while maintaining detection speed. However, the deployment of Transformer-based models to the embedded side is still in the exploratory stage. More exploration is needed to realize the double improvement of ASDINet detection accuracy and speed. In addition to improving the network model, upgrading hardware facilities is also essential. Good lighting design is critical for machine vision systems used in food sorting. In previous research and in this study, the fruit samples were illuminated using a typical lighting technique with the lamp at a 45° angle to the object [55]. However, due to the curvature of the surface of a spherical fruit such as an apple, this type of lighting can create bright spots on the fruit. Direct lighting can easily lead to specular reflections; thus, some researchers paint diffuse white paint on the surface of a custom-made dome or tunnel to maximize reflectivity, thereby providing even diffuse light for uniform illumination [56,57]. Inspired by this, we will refine the light source distribution for the light chamber and apply diffuse white paint to walls in future research. This can improve the detection accuracy of our apple sorting system to a certain extent.

## 6. Conclusions

This study designed an apple surface defect detection network (ASDINet), suitable for less data training from the perspective of specific commercial applications, and conducted practical application tests. The model was validated on images acquired on a low-cost computer vision module consisting of a commercial camera, a robotic arm, a rotating stage, and a homemade LED light. Our network achieved 98.8% AP and a 97.75% F1-score in the comparison experiment with the most advanced model, and the detection speed reached 39ms per frame, achieving a trade-off between accuracy and ease of deployment. These results are better than the traditional image processing method, and are more promising. Loading the ASDINet into the custom software of the fruit sorting system, each category used 25 independent apples to test its online detection performance; the network achieved an accuracy rate of 96%, with a processing time of less than 63 milliseconds per apple. In the data sensitivity experiment, the ASDINet achieved results that met the production needs under the training of 42 defective pictures. These conclusions indicate that the method can reduce manual work, reduce labor costs, and increase the flexibility of the production line when applying this method to new fields. Future research will focus on combining Transformer with the ASDINet, and combining MSI technology to improve lighting design.

## Figures and Tables

**Figure 1 foods-12-01352-f001:**
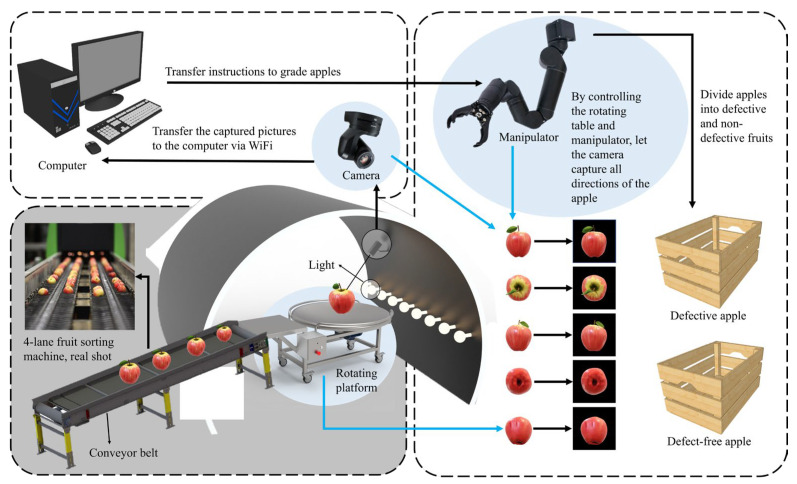
Schematic diagram and physical photos of the optical chamber of the 4-channel fruit sorting machine.

**Figure 2 foods-12-01352-f002:**
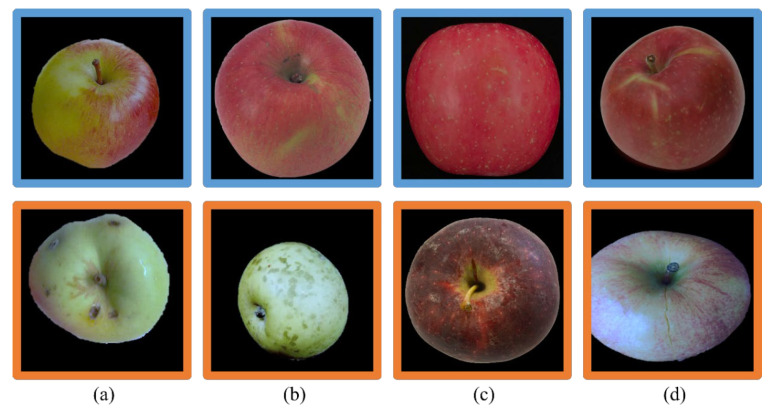
Non-defective apple images with a blue border, and defective apple images with an orange border. Among them, (**a**,**b**) defects are rot and plant disease caused by biological factors, respectively, (**c**) is a laceration caused by environmental factors, and (**d**) is mechanical damage caused during packing and transportation.

**Figure 3 foods-12-01352-f003:**
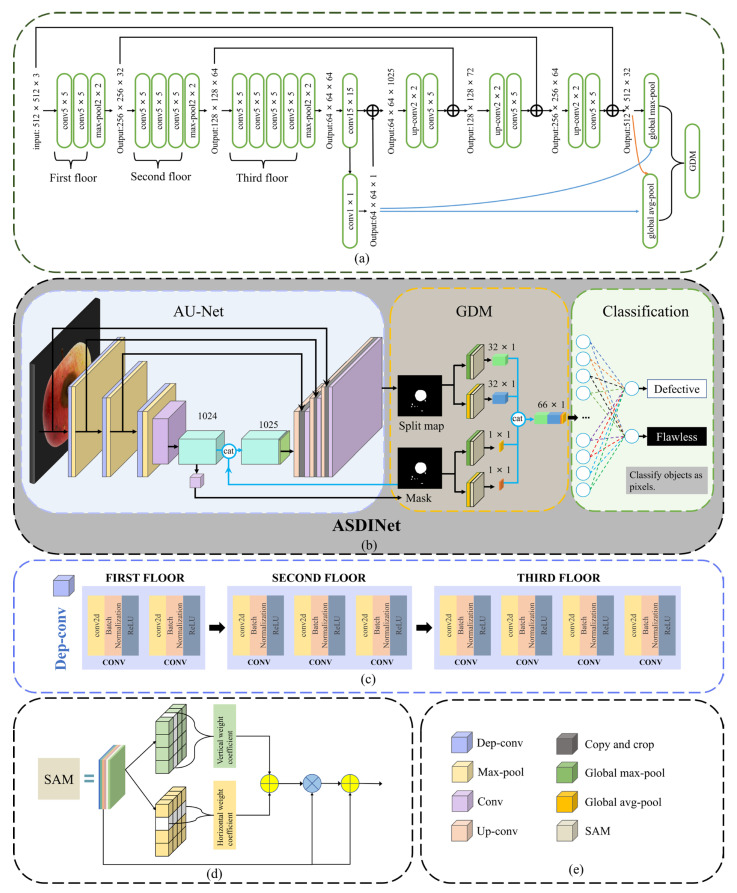
Network structure diagram: (**a**) is the structure diagram of the ASDINet; (**b**) is the network structure diagram of the ASDINet, including the AU-Net and GDM; (**c**) is a schematic diagram of the Dep-conv module; (**d**) is a schematic diagram of the GSAM; (**e**) is the meaning of the corresponding module in (**b**).

**Figure 4 foods-12-01352-f004:**
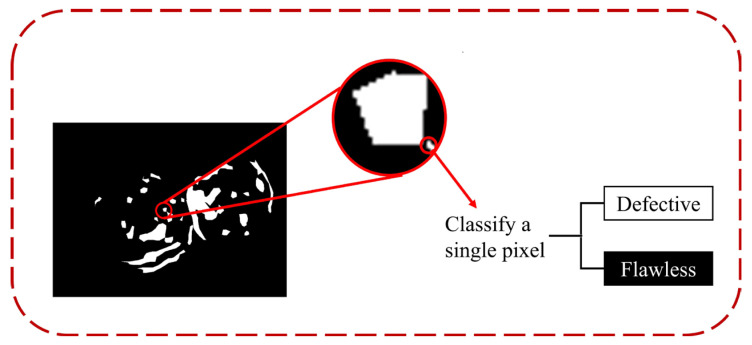
Converting the pixel-level segmentation problem into a binary classification problem and classifying the sample as a single pixel.

**Figure 5 foods-12-01352-f005:**
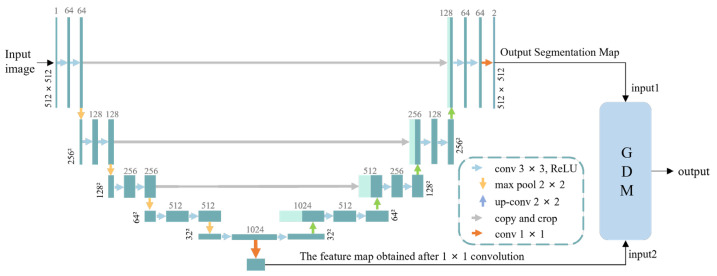
Schematic diagram of the U-Net adding a feature map output. Each dark green box corresponds to a multi-channel feature map. The number of channels is denoted on top of the box. The x-y-size is provided on the lower edge of the box. Light green boxes represent copied feature maps. The arrows denote the different operations.

**Figure 6 foods-12-01352-f006:**
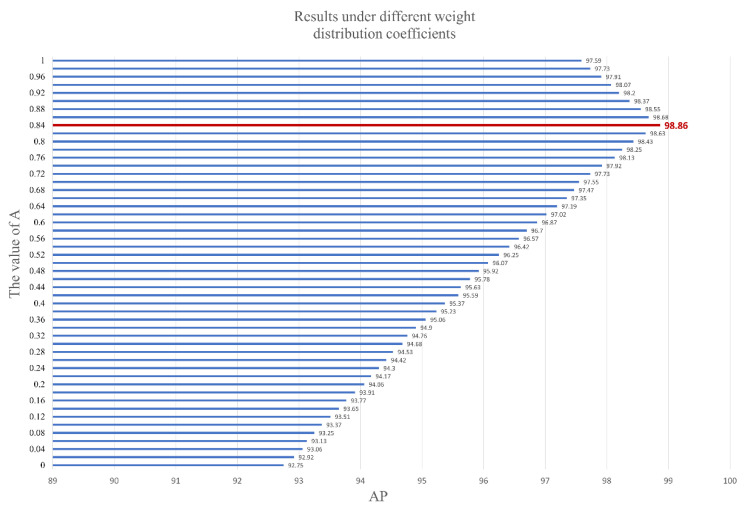
Correspondence between AP and the weight assignment factor. The red line indicates the best result.

**Figure 7 foods-12-01352-f007:**
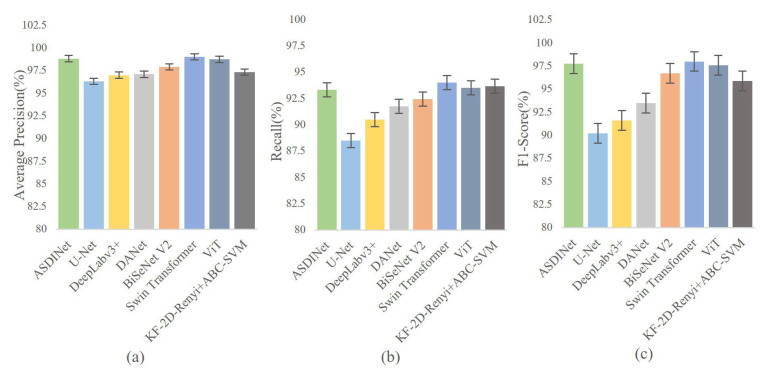
Comparison with state-of-the-art networks. Among them, (**a**) represents the AP of each model, (**b**) represents the Recall of each model, and (**c**) represents the F1-score of each model.

**Figure 8 foods-12-01352-f008:**
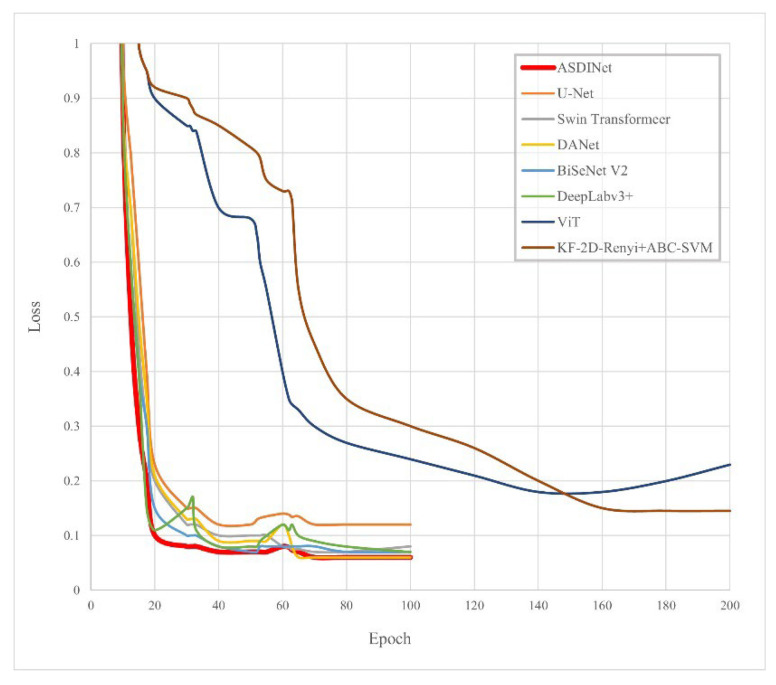
The losses of each model.

**Figure 9 foods-12-01352-f009:**
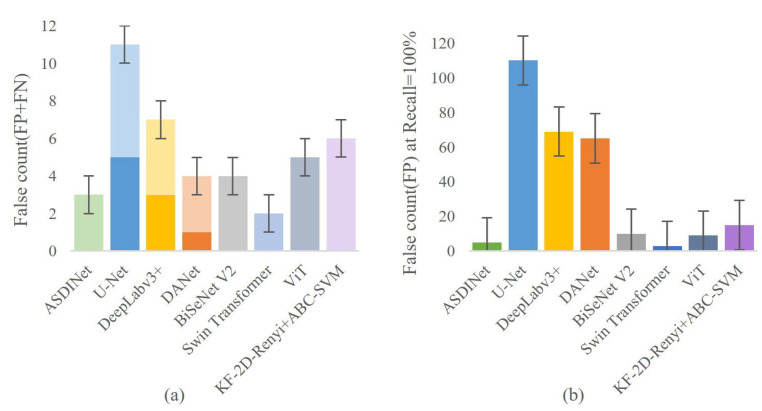
Model performance evaluation (the dark color is FP, and the light color is FN). Among them, (**a**) represents the error count under normal conditions, and (**b**) represents the error count when Recall = 100%.

**Figure 10 foods-12-01352-f010:**
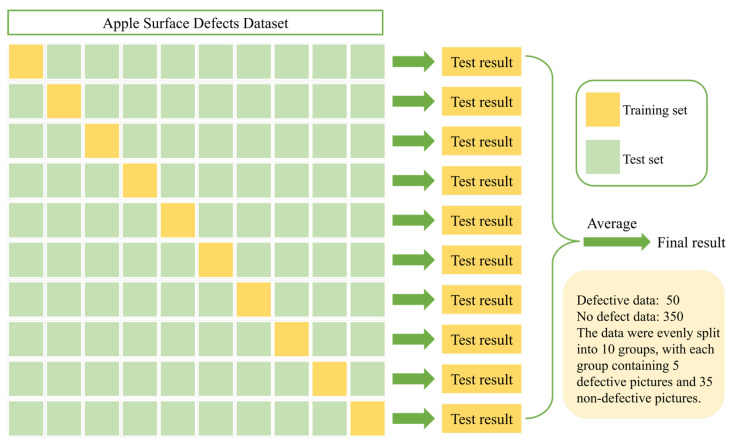
Ten-fold cross-validation.

**Figure 11 foods-12-01352-f011:**
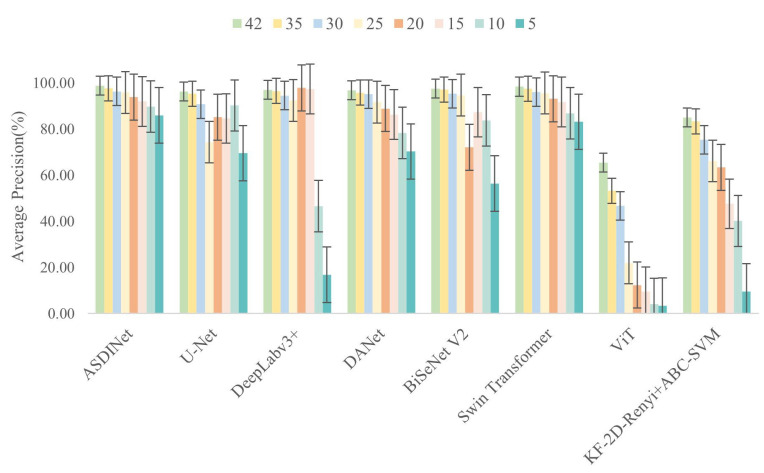
Classification performance of the model with different numbers of positive (defective) training samples.

**Figure 12 foods-12-01352-f012:**
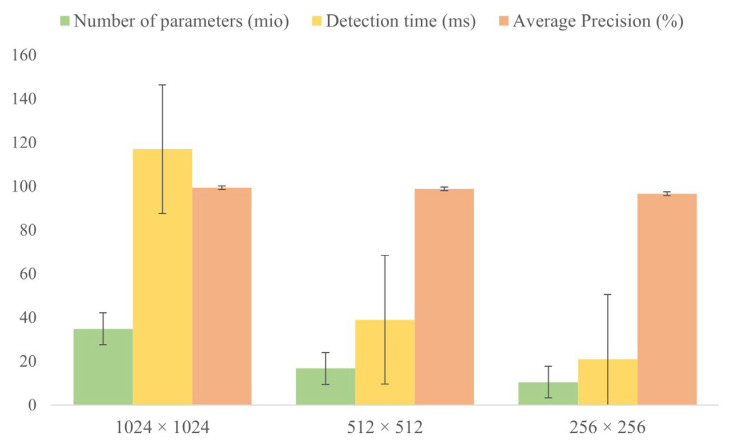
The parameter amount of the model at different image resolutions, the detection time, and the average precision.

**Figure 13 foods-12-01352-f013:**
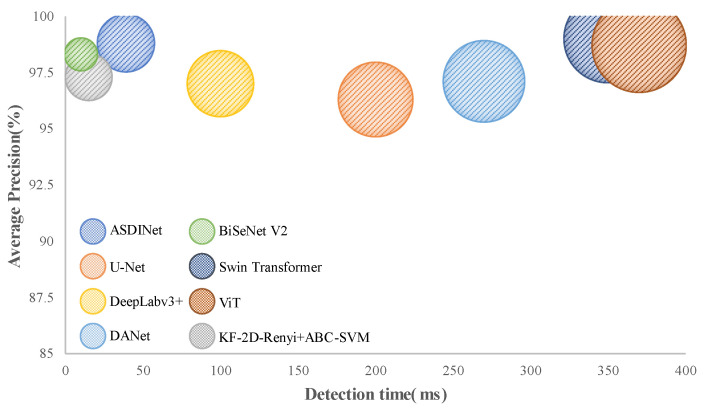
Detection (forward pass) time with respect to the classification performance for a single image.

**Figure 14 foods-12-01352-f014:**
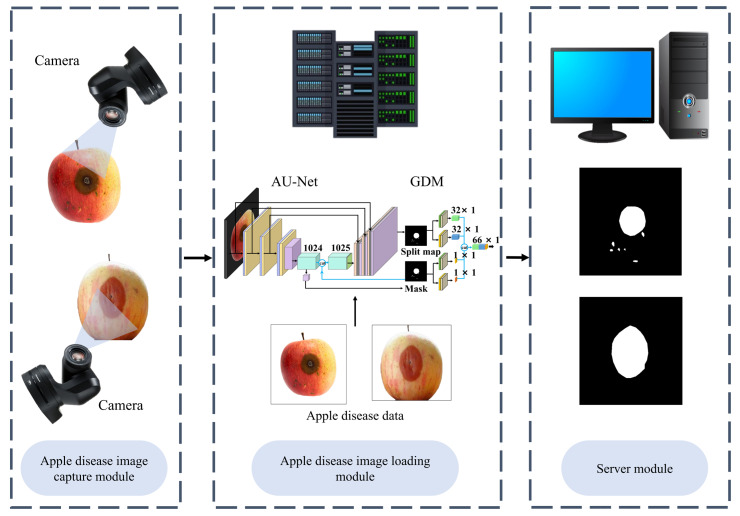
Apple surface defect detection system diagram.

**Figure 15 foods-12-01352-f015:**
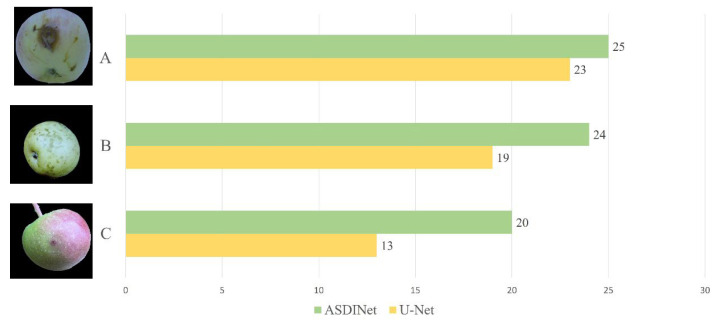
Detection results of three typical apple defects by the ASDINet and U-Net. Out of the three defects, **A** has a moderate size, **B** is relatively large but light in color, and **C** is very small.

**Figure 16 foods-12-01352-f016:**
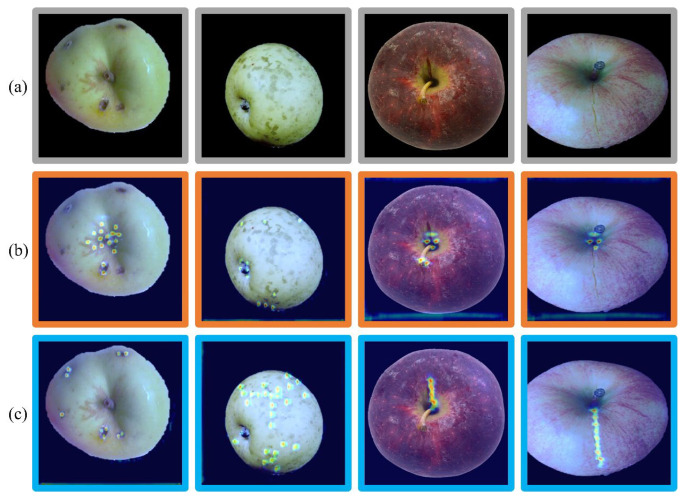
Class activation map (CAM): (**a**) original image, (**b**) ASDINet, (**c**) U-Net.

**Table 1 foods-12-01352-t001:** Related research details.

References	Method	Advantage	Drawback
[32]	Citrus sorting by identification of the most common defects using multispectral computer vision	Without any training, the sorting accuracy can reach as high as 95%	Cannot distinguish stem ends from defects
[33]	Mechatronic components in apple sorting machines with computer vision	Reduced the false detection rate of fruit stems and calyxes	Defects in the edge portion of the fruit are difficult to detect
[34]	Deep learning implementation using convolutional neural network in mangosteen surface defect detection	The classification accuracy of mangosteen surface defect detection can reach 97%	The model is simple and cannot fully mine the feature information of the data
[35]	Improved classification approach for fruits and vegetables freshness based on deep learning	The system not only identifies the type of fruit or vegetable, but also classifies it as fresh or rotten	Misclassification occurs when fruits and vegetables have very similar external features
[36]	Pre-trained OverFeat network	Alleviates data requirements	This approach does not exploit the full potential of deep learning, as it does not learn the network on the target domain
[37]	Compact convolutional neural network for textured surface anomaly detection	Defects can be clearly segmented, and the existence of defects can be classified for each image	This approach does not exploit the full potential of deep learning, as it does not learn the network on the target domain

**Table 2 foods-12-01352-t002:** The data distribution of the training set.

Category	Number	Proportion (%)
Normal	396	79.2
Rot	31	6.2
Disease	21	4.2
Laceration	27	5.4
Mechanical injury	25	5.0

**Table 3 foods-12-01352-t003:** Hardware and software parameters.

Hardware environment	CPU	12th Gen Intel(R) Core (TM) i7-12700H 2.30 GHz
	RAM	32 GB
	Video memory	32 GB
	GPU	NVIDIA GeForce RTX 3070 Ti Laptop GPU
Software environment	OS	Windows 11
	CUDA Toolkit V11.3 CUDNN V8.0.4 Python 3.8.8 torch 1.8.1; torchvision 0.9.1	

**Table 4 foods-12-01352-t004:** Experimental settings.

Image Size	Batch_Size	Learning Rate	Decay	Iterations	Loss Function
512 × 512	1	0.0005	0.0005	100 epochs	Cross-entropy

**Table 5 foods-12-01352-t005:** Comparing the impact of the U-Net and AU-Net on the final benefit of the network. In order to rule out that our experimental data were obtained by random or accidental occurrence, we used SPSS to conduct a one-way analysis of variance on the experimental data. Our null hypothesis ($H_{0}$) was that performance metrics were equal, and any small gains or losses observed were not statistically significant. $H_{1}$ was the alternative hypothesis. We set α=0.05. The analysis results show that F=5.764>F crit=5.318, P=0.043<α=0.05, reject $H_{0}$. We performed the same analysis on the data in Table 6 and Table 7; please refer to the following for details.

Methods	Accuracy	Precision	Recall	AP	F1-Score
U-Net	95.57	94.31	88.52	96.32	90.20
AU-Net	98.86	98.53	93.32	98.83	97.75

**Table 8 foods-12-01352-t008:** Learning settings and hyperparameters.

Method	Image Size	Batch_Size	Learning Rate	Decay	Iterations	Loss Function
ASDINet	512 × 512	1	0.1	0.0005	100 epochs	Cross-entropy
U-Net	512 × 512	1	0.1	0.0005	100 epochs	Cross-entropy
DeepLabv3+	512 × 512	1	0.1	0.0005	100 epochs	Cross-entropy
DANet	512 × 512	1	0.1	0.005	100 epochs	Cross-entropy
BiSeNet V2	512 × 512	1	0.1	0.0005	100 epochs	Cross-entropy
Swin Transformer	512 × 512	1	0.1	0.0005	100 epochs	Cross-entropy
ViT	512 × 512	1	0.1	0.0005	200 epochs	Cross-entropy
KF-2D-Renyi+ABC-SVM	512 × 512	1	0.1	0.0005	200 epochs	Cross-entropy

**Table 9 foods-12-01352-t009:** The performance of the ASDINet trained between 50 and 5 defective images.

Model	50	45	40	35	30	25	20	15	10	5
ASDINet	98.83	98.52	98.10	97.72	96.33	95.91	93.84	92.01	89.80	85.96

**Table 10 foods-12-01352-t010:** The performance of the ASDINet trained between 45-40 defective images.

Model	45	44	43	42	41	40
ASDINet	98.83	98.83	98.83	98.83	98.65	98.57

**Table 11 foods-12-01352-t011:** Results and the average of 10 times 10-fold cross-validation.

Times	1	2	3	4	5	6	7	8	9	10
Accuracy	98.81	98.80	98.84	98.83	98.83	98.81	98.85	98.83	98.83	98.83
Average	98.826									

**Table 12 foods-12-01352-t012:** Comparison with the state-of-the-art methods in the number of learnable parameters and average precision.

Method	Number of Parameters	Average Precision (%)
U-Net	31.1 mio	96.32
DeepLabv3+	41.3 mio	97.00
DANet	63.4 mio	97.11
BiSeNet V2	15.3 mio	97.92
Swin Transformer	106.2 mio	99.03
ViT	117.6 mio	98.75
KF-2D-Renyi+ABC-SVM	15.6 mio	97.34
ASDINet	16.7 mio	98.83

**Table 13 foods-12-01352-t013:** The average detection time of U-Net and ASDINet in the three cases of A, B, and C.

Defect Category	ASDINet	U-Net
A	41 ms	199 ms
B	48 ms	253 ms
C	63 ms	418 ms

## Data Availability

The original contributions presented in this study are included in the article; further inquiries can be directed to the corresponding author.

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
