# Peer review of "Automatic Detection of Small Sample Apple Surface Defects Using ASDINet"

_foods, 2023, doi:10.3390/foods12061352_

Round 1

Reviewer 1 Report

Dear authors,

The proposed article is methodologically correct but has some major flaws, the major one being the total absence of statistical validation of the experimental data. Other requested improvements can be found in the specific comments.

In general, the results part needs to be simplified and better organised.

Another aspect to be pointed out is the issue of sample size. In my opinion, the number of samples submitted for analysis is low. A statistical robustness for data validation, especially using the techniques reported in the article, requires a much larger number of data.

Specific comments:

Line 126: Please specify better the speed unit. Author means apple/s?

Line 136 – 137: “A background-removed apple image was obtained by thresholding the red component of the RGB image using Otsu”. The sentence reports you used the Otsu method (insert please the reference), to threshold the images on the basis of red-channel. Figure 1 depict images of apple having red and green color. Thus, it’s too hard/not possible threshold a green apple using Otsu method on the red channel. Please improve the sentence indicating the right way to threshold all the different images.

Table 2. Insert please the legend of symbols used in the equation. Moreover, the equation 6 seems to be too cryptic. If possible, simplify the terms used in a clearer way.

Line 209. I suggest inserting a reference for ReLU function used.

Line 227-228: “In digital image processing, the mask is 227 a two-dimensional matrix array or a multivalued image”. The sentence is incorrect: the mask represents a “logic image” or 2-bit image consisting of a matrix having elements characterized by 2 only values: 0 or 1. The multivalued image refers to other type of images.

Line 235: Pleas define GDM acronym.

Line 339: Did you consider the minimum detail size detectability by reducing the image size to 512X512? Moreover, which was the color depth of the captured images?

Figure 5. I suggest to describe some features of figure in the caption.

Tables 5, 6, 7. Insert the statistical significance..

Author Response

Dear Reviewer

I would like to express my sincere gratitude for taking the time out of your busy schedule to review our manuscript. Your insightful and thoughtful comments have been extremely helpful in improving the quality of our work. We greatly appreciate your effort and expertise in providing us with such valuable feedback.

We have carefully considered your comments and suggestions, and have made the necessary revisions to our manuscript. Your constructive feedback has helped us to refine our research and to present our findings in a more clear and coherent manner.

We have included our specific response to your comments in the Word. We hope that you find our revisions satisfactory and that our work meets the high standards of your esteemed journal.

Once again, thank you very much for your time and effort in reviewing our manuscript. Your contribution has been invaluable to our research.

Best regards,
The authors

Author Response

(The authors gave the same response as above.)

Round 2

Reviewer 1 Report

Dear authors,

The manuscript have been improved following the suggestions. Then, I suggest to consider the paper for publication.

Author Response

The revised manuscript is attached below.
